# Relationship of Self Efficacy in Medication Understanding with Quality of Life among Elderly with Type 2 Diabetes Mellitus on Polypharmacy in Malaysia

**DOI:** 10.3390/ijerph19053031

**Published:** 2022-03-04

**Authors:** Nur Athirah Rosli, Md Yasin Mazapuspavina, Zaliha Ismail, Nahlah Elkudssiah Ismail

**Affiliations:** 1Department of Primary Care Medicine, Faculty of Medicine, Univerisiti Teknologi MARA, Selayang Campus, Batu Caves 68100, Selangor, Malaysia; trahrosli@yahoo.com; 2Department of Public Health Medicine, Faculty of Medicine, Universiti Teknologi MARA, Sungai Buloh Campus, Sungai Buloh 47000, Selangor, Malaysia; zaliha78@uitm.edu.my; 3Clinical Pharmaceutics Research Unit, MAHSA University, Bandar Saujana Putra, Jenjarom 42610, Selangor, Malaysia; elkudssiah77@yahoo.com; 4Malaysian Academy of Pharmacy, Puchong 47120, Selangor, Malaysia

**Keywords:** elderly/geriatric, type 2 diabetes mellitus, medication understanding, self-efficacy, quality of life

## Abstract

Self-efficacy (SE) has been shown to be positively correlated with quality of life (QOL) among patients with type 2 diabetes mellitus (T2DM). Medication understanding (MU) on the other hand, leads to good adherence that indirectly improves QOL. Measuring self-efficacy in medication understanding is useful to ascertain patient’s confidence in medication adherence. However, there is a lack of studies on the relationship between self-efficacy in medication understanding with QOL. This study aimed to determine the relationship between self-efficacy in medication understanding and QOL, and the factors associated with QOL in elderly with T2DM on polypharmacy. A cross-sectional study was conducted on these populations at primary care specialist clinic. Malay version of MU in SE questionnaire (MUSE) was used. Higher scores showed a better understanding. A revised Version Diabetic Quality of Life-13 (RVDQOL-13) questionnaire was used with lower scores indicating higher QOL. A total of 321 patients participated, with the majority being male (58.3%), Malay (84.7%), a predominant age group of 60–69 (75.7%) with mean age (±SD) of 66.7 (±0.286) years old. The median (IQR) of MUSE was high—30 (4)—while the RVDQOL-13 was low—19 (8)—which demonstrated high QOL. Inverse correlation was found between MUSE and QOL (*r* −0.14, *p* < 0.01). Multiple linear regression analysis demonstrated that MUSE score (β −0.282; 95% CI: (−5.438, −2.581); *p* < 0.001), low-income group (β −0.144; 95% CI: (−3.118, −0.534); *p* = 0.006) and duration of medications ≥240 days (β −0.282; 95% CI: (−5.438, −2.581); *p* < 0.001) were associated with better QOL, while medications ≥10 (β 0.109; 95% CI: 0.214, 4.462; *p* = 0.031) and those with pills and insulin (β 0.193; 95% CI: 1.206, 3.747; *p* < 0.001) were associated with poor QOL. In conclusion, higher MUSE is associated with better QOL. Findings suggest emphasizing self-efficacy in medication understanding in the management of elderly with T2DM on polypharmacy to improve QOL.

## 1. Introduction

According to the United Nations, the elderly population is defined as people who are 60 years or older, and it is predicted to expand by 56% globally between 2015 and 2030, with the most rapid growth in cities of developing countries [1]. The global population of those aged 60 years were over 382 million in 1980 and increase to 962 million in 2017 worldwide. In Malaysia, the elderly population prevalence is expected to increase from 9.2% in 2015 to 23.6% by year 2050. This is expected to put a significant pressure on the country’s healthcare systems. [2].

The Malaysian National Health and Morbidity Survey (NHMS) 2015 observed that the disease trend among the elderly has shifted from aging-related diseases to lifestyle-related diseases, such as, hypertension, hypercholesterolemia, chronic obstructive pulmonary disease, and Type 2 Diabetes Mellitus (T2DM) [2]. T2DM is becoming more common worldwide due to increased life expectancy and lifestyle changes [3]. Globally, the number is expected to increase from 220 million in 2010 to 300 million by 2025. In Malaysia, according to the latest report from NHMS, showed an increasing prevalence trend of patients with T2DM, from 13.4% in 2015 to 18.3% in 2019 and it is increasing with age. The prevalence was reported to peak at 34.3% among people aged 65 to 69 years old [4].

The elderly is in poorer health than the youth, as the debilitating effects of numerous diseases are compounded with the physical and social changes associated with ageing. Hence, an increase in the proportion of elderly is associated with an increase in the prevalence of diseases [5]. More than 60% of adults will have two or more chronic diseases by the age of 65, a condition known as multimorbidity, while more than 25% will have four or more chronic diseases and over 10% will have six or more chronic diseases [6]. The presence of multimorbidity increases the complexity of therapeutic management and it has a detrimental impact on health outcomes [7]. Elderly with T2DM pose a particular challenge to clinicians concerning managing medications as they are prescribed more medicines and are exposed to more drug interactions than their non-T2DM counterparts. Lim et al. supported that the number of medications used by elderly with endocrine diseases was higher than those with cardiovascular diseases [8]. Multiple drugs not only increase the cost and complexity of therapeutic regimens, but also increase the risk of adverse drug reactions [7]. While nonpharmacological approaches for managing diabetes and its complications are important, medication remains the cornerstone of management. As a result, there are a number of strong factors that favor multiple drugs or polypharmacy in diabetic patients [7].

Polypharmacy is described as the use of five or more drugs. Other definitions of polypharmacy involve duration of therapy either 90 days to 239 days or ≥240 days or more (‘long term use’) [9]. Polypharmacy was found to be 45.9% common among urban community-dwelling elderly in Malaysia, with 576 people out of 1256 are using at least five drugs. In total, 499 (86.6%) of those with polypharmacy exposure received five to nine medications, 65 (11.3%) received 10 to 14 medications, and 12 (2.0%) received 15 or more [8]. Physiological changes in the ageing kidney, memory problems and different treatment regimens all add to the difficulty of therapy [5]. It raises the likelihood of adverse drug reactions through altering medication absorption, distribution, metabolism and excretion [10].

There is an increase in the patient’s frailty linked to the number of their medicines as frailty increases by 1.5 times with five or more drugs [11]. Geriatric syndromes are common and they can be caused by using the inappropriate drugs [12]. Even mild cognitive impairment might lead to care errors that can result in significant consequences [13]. Elderly individuals with T2DM are also susceptible to orthostatic hypotension, insomnia and constipation with commonly used drugs, such as opioids, anticholinergic agents and αblockers [7]. Thus, it is vital to develop strategies to limit the prescription of unnecessary medications in order to improve the QOL of diabetic patients. [14]. Most studies found that people with diabetes have a lower QOL than those without diabetes [15]. Thus, understanding and compliance regarding medication are critical for avoiding drug-related issues and subsequently leading to better QOL [16].

Medication understanding (MU) is described as how patients feel, react and think about their medications [17]. Patients’ misunderstanding of prescription medication instructions has been recognized as health literacy issue and factor affecting patient’s safety [18]. Patients frequently misunderstand the correct dose of a prescription as well as the warnings and precautions associated with the medication [19]. Medication errors frequently result from patients’ unintentional misuse of prescribed medication. Hence, there has been an emphasis on improving provider-patient communication around the topic of medication use. Simplifying prescription regimens, decreasing pill burdens and providing better explanations of the needs for the medications should be targeted for intervention to increase MU in the elderly. Among other factors, health literacy and self-efficacy (SE) have been repeatedly identified as predictors of one’s ability to understand medication instructions [19].

Bandura (1986) defined SE as people’s judgements of their capabilities to organize and execute courses of action required to attain designated types of performances. SE is a significant indicator of adherence in all types of diseases [20]. Multi-morbid primary care patients with lower SE and higher disease load have lower QOL [21]. Awareness of SE levels among individuals with multi-morbidity may aid health professionals identify patients who require additional self-management support. Providing chronic disease self-management support has been hailed as a hallmark of good care. Higher SE may lead to enhanced QOL in multi-morbidity and promote positive behavioral health [21,22]. Low SE will lead to a decline in an individual’s differential association, differential reinforcement, imitation and definition. A decline in these social parameters will further lead to a decline in medications adherence and SE [23]. Higher SE was linked to better glycemic control, medication adherence, self-care and mental health-related QOL [24].

Research looking into the association of MU and SE had been conducted in other countries with various findings reported. Potentially, a study that is inclusive of factors known to impact diabetic QOL as a result of poor MU and SE may provide better knowledge on how MU and SE may affect diabetic QOL in the elderly. Bowen et al. also reported a relationship between QOL and SE. They propose that person with higher SE dealing with their diabetes had better QOL, and findings are in line with previous studies on the impact of SE on QOL [25]. QOL is a significant health outcome in its own right, serving as the ultimate goal of all health interventions. Most studies found that persons with diabetes have a lower QOL than people without diabetes, particularly in terms of physical functioning and wellbeing. QOL measures should be used to coordinate and evaluate treatment interventions [15].

Patients with T2DM have been shown to have a positive correlation between SE and QOL. MU, on the other hand, leads to better adherence, which indirectly improves QOL. In measuring SE in MU, it may be useful to ascertain changes in patient’s confidence related to medication use [19]. To our knowledge, no research has been completed on the relationship between SE in MU or MUSE with QOL in elderly with T2DM on polypharmacy. Therefore, the objective of this study was to determine the level of SE in MU and QOL, the correlation between SE in MU and QOL and factors associated with QOL in elderly with T2DM on polypharmacy.

## 2. Materials and Methods

### 2.1. Study Design and Setting

A cross-sectional study was carried out at an institutional Primary Care Specialist Clinic (PCSC) in Selangor from December 2019 to November 2020. Services provided here include walk-in clinics for health screening, acute ailments, and appointment-based clinics for follow-up of chronic diseases. In addition, the clinic has access to radiological imaging, laboratory and referral services to other specialties. An average of 80 patients attending this clinic per day. We selected this clinic because it was located in an urban area with heavy patients’ load, including elderly patients with T2DM. This PCSC was run by family medicine specialists, consultants and postgraduate doctors pursuing a Master’s degree in family medicine from the institution. The vast majority of the patients are from Klang Valley.

### 2.2. Sampling Frame

The study population were elderly patients with T2DM on polypharmacy who received care at the institutional PCSC. The inclusion criteria were patients aged ≥60 years old, intact cognitive function, have been diagnosed with T2DM for at least one-year duration, received follow up care at institutional PCSC for at least once within the last one year, on polypharmacy which is ≥5 medications and was able to read and communicate in either Malay or English language.

### 2.3. Sampling Method

Convenience sampling of elderly who attended the institutional PCSC for follow-up were screened for eligibility through their medical record system. The researcher approached and explained to patients while waiting for their clinic consultation. Later, the researcher invited those eligible to participate in the study using inclusion and exclusion criteria.

### 2.4. Study Tools

Mini-Cog is a brief measure for assessing cognitive impairment in our participants as one of the inclusion criteria is intact cognitive function. It was developed and validated by Borson et al. with a sensitivity of 99% and specificity of 93%. It consists of a 3-item recall (1 point for each word) and clock drawing (2 points for a normal clock). The 3-item recall and clock drawing scores together with a total score of ≥3 indicate a lower likelihood of dementia [26].

A standardized form was used to collect sociodemographic and clinical data information. Part A consisted of age, gender, ethnic group, marital status, educational level and household income. Part B: clinical data included the number of prescribed medications, duration of T2DM, other co-morbidities and modalities of treatment, whether it is ‘pills’, a mix of ‘pills and insulin’ or ‘combined with other modalities’ such as Meter Dose Inhaler (MDI). Part C consisted of two sets of questionnaires that had been translated and validated: The Self Efficacy in Medication Understanding (MUSE) Malay version (27) and the Revised Version of Diabetic Quality of Life (RVDQOL-13) Malay version (28).

The MUSE is a brief, valid and reliable research questionnaire which can be used in clinical practice and research to evaluate patients’ understanding and use of prescription medication. The MUSE provides a more general approach to measuring self-efficacy in medication use than existing disease- or context-specific measures. MUSE assesses self-efficacy in medication use and it also emphasizing the importance of patient medication understanding. This scale was developed originally from two subscales of the CASE-Cancer measure [27]; these subscales were supplemented with additional items intended to reflect participants’ understanding of and confidence in taking their prescription medications. The MUSE Malay version [28] was chosen to measure this study population’s understanding and SE levels because of the language used and its good reliability (Cronbach’s α of 0.89). The self-administered questionnaire consisted of an 8-item scale to assess patient’s self-efficacy in learning about and taking medications with a four-point Likert scale, with 1 = strongly disagree, 2 = slightly disagree, 3 = slightly agree, and 4 = strongly agree. Overall, the scores ranged from 8 to 32, with higher scores indicating greater levels of medication understanding.

The RVDQOL-13 Malay version is a self-administered questionnaire comprising 13 items with three domains measuring diabetic patients’ QOL (DQOL) [29]. The three domains include ‘satisfaction’, ‘impact’ and ‘worry’. The Malay version of RVDQOL-13 has good composite reliability for each domain; “satisfaction” domain showed highest composite reliability of 0.922, followed by “worry” domain (0.794) and “impact” domain (0.781). Response choices of satisfaction are scored on a five-point Likert scale, with responses of 1 = very satisfied, 2 = moderately satisfied, 3 = neither satisfied nor dissatisfied, 4 = moderately dissatisfied and 5 = very dissatisfied in with a range score from 6 to 30, worry domain are scored from 1 = never, 2 = sometimes, 3 = often, 4 = frequently and “impact” domain are scored from 5 = always with range score from 4 to 20 and 3 to 15, respectively, giving a total score ranging from 13 to 65. The author proposed the score for each domain and the total score to be converted to percentage. Higher total scores indicate poorer quality of life.

### 2.5. Sample Size Calculation

Few sample sizes were calculated based on a few different prevalence according to the objectives. All the sample sizes were calculated using the Single Proportion formula based on the study’s objective. As a conclusion, the highest sample size was taken according to the variation in medication understanding among elderly (62%) by Spiers et al. [30]. The confidence interval was taken as 95%, power 80%; the minimum sample required for the study was 321 patients. Considering a 10% non-response and non-eligible rate, this study aimed to approach at least 353 participants.

### 2.6. Data Collection and Study Procedure

The patients who attended their follow-up were screened for eligibility using the medical record system. The researcher approached eligible patients in the waiting area after they had their registration numbers. They were given a patient information leaflet outlining the study and its objectives. Consent was then obtained from those who were interested in participating and met the inclusion and exclusion criteria, which included intact cognitive function as measured by the Mini-Cog. Only one investigator was trained and involved in the study procedures before the conduct of the study to minimize variability in the method of data collection.

### 2.7. Questionnaires Administration

Participants were given a set of questionnaires containing sociodemographic and clinical characteristics, MUSE Malay and RVDQOL-13 Malay versions. Both verbal and written instructions were given on how to complete the questionnaires. They were asked to circle or mark the options that best suited them. Should any queries arise, participants were encouraged to seek clarification from the investigator at any time. Participants took an average of 10–20 min to complete the questionnaires. They handed the questionnaires to the researcher once they were finished, who double-checked the answers for completeness. Figure 1 illustrates the conduct of the study.

### 2.8. Definition of Terms

The elderly was divided into three age groups. The young-old (60–69 years old), middle-old (70–79 years old), and very-old (over 80 years old). According to the Malaysian educational system, education levels are classified as follows: no formal education, primary school (standard 1–6; ages 7–12), secondary school (form 1–5; age 13–17), and tertiary education (college or university) [31]. Household incomes were divided into three categories: low (B40) with a monthly household income of <RM 4850, middle (M40) with income between RM 4,850 and RM 10, 959, and high (T20) with a monthly household income of >RM 10960 [32].

### 2.9. Statistical Analyses

Data were analyzed using the Statistical Package for the Social Science (SPSS) version 26.0 (IBM). All continuous variables were described as median (IQR) and number (n) and percentages (%) for dichotomous or nominal data. The levels of MUSE and DQOL were analyzed using median (IQR) as the data were not normally distributed. Relationships between MUSE and DQOL were analyzed using Spearman’s correlation test. Factors associated with DQOL amongst the study population were analyzed by simple linear regression (SLR) followed by multiple linear regression (MLR). Variables with a *p* value of less than 0.25 by SLR were then included in the MLR. A *p* value of less than 0.05 was considered statistically significant in the MLR.

## 3. Results

A total of 360 elderly patients with T2DM were screened earlier from the system and were approached consecutively and invited to enter the study. Out of this, 27 (7.5%) were not eligible to enter the study as they did not fulfil the inclusion and/or criteria and 12 patients (3.3%) refused to participate. Therefore, the study’s recruitment rate was 89%, resulting in 321 eligible T2DM participants who completed the questionnaires.

### 3.1. Characteristics of the Study Population

The sociodemographic and clinical characteristics of the participants are shown in Table 1. Majority of the participants (58.3%) were male with mean age (±SD) of 66.7 (±0.286) years old. The participants were ethnically diverse, comprising (84.7%) Malays, (6.9%) Chinese, (8.1%) Indians and (0.3%) one participant from Sri Lanka. Most of the respondents were married (80.7%), educated up to the secondary school level (85.7%) and regarding household income, 64.5% were from the low-income group (B40).

### 3.2. Levels of MUSE and RVDQOL-13

The used MUSE and RVDQOL-13 have good reliability or internal consistency among our participants with Cronbach’s alpha of 0.867 and 0.846, respectively. Table 2 and Table 3 show the total median score of MUSE and RVDQOL-13 of the participants. The total median scores (IQR) for MUSE were 30 (4) that reflect a high median MUSE score and RVDQOL-13 of 19 (8) that reflect a high level of diabetic quality of life, respectively.

Table 2 illustrates the median scores (IQR) for each MUSE subscale domain and item. Both subscale domain for ‘taking medication’ and ‘learning about medication’ have median scores (IQR) of 16 (2) and 15 (2), respectively.

Table 3 shows the median subscale scores for RVDQOL-13 subscale domains among participants with T2DM. The highest median subscale domain score of 5 (3) was for ‘worry domain’ followed by 6 (4) for ‘impact domain’ and the lowest median subscale domain score of 8 (3) was for ‘satisfaction domain’. It was also reported in percentage as suggested by the RVDQOL-13 original author. Participants score the best in ‘satisfaction’ domain with 26.67%, followed by 30% in ‘impact’ domain and 33.33% in ‘worry’ domain. The overall total score was 29.23% in which reflected the lower the score, the better the QOL.

### 3.3. Relationships between MUSE and RVDQOL-13

Table 4 shows the relationship discovered between taking ‘medication and satisfaction domains’ (*r* −0.242, *p* <0.01, ß −0.133, *p* < 0.01), ‘taking medication and impact domains’ (*r* −0.136, *p* < 0.015, ß −0.154, *p* < 0.006), and ‘learning medications and satisfaction domains’ (*r* −0.228, *p* < 0.01, ß −0.143, *p* < 0.010). Overall, there was a weak negative correlation between total MUSE and RV-DQOL13 (*r* −0.14, *p* < 0.001, ß −0.186, *p* < 0.001), reflected that participants with high MUSE has better QOL. Figure 2 illustrates the correlation between MUSE and RVDQOL-13.

### 3.4. Factors Associated with QOL

The findings of the univariate analysis using SLR are shown in Table 5. The independent variables that were entered into the SLR for QOL were middle-old age group, Chinese ethnicity, married, divorce, private sector employment, housewife, pensioner, B40, medications ≥ 10, duration of medications ≥240 days, cardiovascular-related disease, respiratory-related disease, other related disease, those who are on pills and insulin, domain taking medication score and total MUSE score. All of these variables had a significant *p*-value of <0.25 and were included in the MLR model.

Table 6 presents the final model of multiple regression analysis in determining the factors associated with QOL. In this model the total MUSE score was used. There were five variables with *p* < 0.05 which accounted for 21.2% (coefficient of determination, *R*² = 0.212) of the variation in DQOL between individuals, medications ≥10 (β 0.109; 95%CI: 0.214, 4.462; *p* = 0.031) and on pills and insulin (β 0.193; 95%CI: 1.206, 3.747; *p* < 0.001), were positively correlated with DQOL in which they have poorer QOL, while low-income group (B40) (β −0.144; 95%CI: −3.118, −0.534; *p* = 0.006), duration of medications ≥240 days (β −0.282; 95%CI: −5.438, −2.581; *p* < 0.001) and total MUSE score (β −0.282; 95%CI: −5.438, −2.581; *p* < 0.001) were negatively correlated with DQOL, in which they have better QOL. The analysis demonstrated that the low-income group (B40) with a medication duration of ≥240 days and higher SE in MU has better QOL, while those on ≥10 medications, as well as pills and insulin, have poorer QOL. Those with better SE in MU were five times more likely to have better QOL.

Table 7 presents the final model of multiple regression analysis in determining the factors associated with QOL. In this model, the subgroup analysis of MUSE score (domain ‘taking medication’ and ‘learning about medication’) were used. There were five variables with *p* < 0.05 which accounted for 18.6% (coefficient of determination, *R*² = 0.186) of the variation in QOL between individuals, medications ≥10 (β 0.113; 95%CI: 0.273, 4.572; *p* = 0.027) and on pills and insulin (β 0.190; 95%CI: 1.152, 3.723; *p* < 0.001), were positively correlated with DQOL in which they have poorer QOL, while low-income group (B40) (β −0.135; 95%CI: −2.997, −0.420; *p* = 0.010), duration of medications ≥240 days (β −0.275; 95%CI: -5.365, −2.469; *p* < 0.001) and domain MUSE of taking medication (β −0.193; 95%CI: −0.852, −0.270; *p* < 0.001) were negatively correlated with QOL, in which they have better QOL. The analysis revealed that the low-income group (B40) with a medication duration of ≥240 days and higher self-efficacy in taking medications has better QOL, while those on ≥10 medications, as well as pills and insulin, have poorer QOL.

## 4. Discussion

To the best of our knowledge, this was the first study evaluating the relationship of MUSE with QOL in elderly with T2DM on polypharmacy in Malaysian primary care setting.

### 4.1. Levels of MUSE and DQOL

This study demonstrated a high median MUSE score (30). In addition, participants were found to have a high, near-maximal total median score indicating a good response for taking medication and learning about medications. According to social cognitive theory, individuals with higher self-efficacy are more likely to carry out a specific behavior, such as taking medication [33].

This result is comparable with the study by Tharek et al., which demonstrated a moderately high median SE and participants were most efficacious in tasks relating to medication intake [34]. Furthermore, Sharoni et al. found that patients had a high level of self-efficacy, and that they are most confident about tasks related to medication, such as taking it as prescribed and maintaining medication intake [35].

A low level of QOL was found among participants in our study that reflected better QOL and participants reported that they were satisfied with their QOL. Shooka et al. concluded that patients with T2DM in Iran reported a modest QOL [36]. However, the findings of this study contradict those of a study conducted in a diabetic clinic in Malaysia, which found that diabetes was linked to a lower QOL [37].

### 4.2. Relationships between MUSE with DQOL

A negative correlation between MUSE and DQOL was demonstrated in elderly patients with T2DM on polypharmacy, particularly between (taking medication and satisfaction domains) and (taking medication and impact domains) in which those with better SE in taking medication has better satisfaction and impact on QOL. Meanwhile, those with better SE in learning medication have better satisfaction in QOL. Overall, MUSE correlated with improved QOL. This is further supported by Peters et al. and Bowen et al., suggesting that elderly individuals with higher SE have a better QOL [21,25].

### 4.3. Factors Associated with QOL

Almost all of our participants have good SE in MU which correlates with better QOL. At this PCSC, all of the doctors were specialists in family medicine or postgraduate doctors pursuing a Master’s degree in family medicine. Their ongoing clinical training and professional development may have an impact on these patients’ MUSE as they may have used interactive communication between patients during medical consultations to ensure adequate understanding. When comparing self-efficacy in learning about medications to self-efficacy in taking medications, participants with higher self-efficacy in taking medications had higher QOL.

The QOL score for patients on pills and insulin was lower than patients who did not use insulin, and Redekop et al. found a similar finding in their study [38]. We found that for every one medication increment in those who are on pills and insulin, or those who are on ≥10 medications, the QOL will decrease by 21.3% and 11.3%, respectively. This finding was consistent with Alonso et al., who identified a relationship between taking more medications and having a lower QOL [39].

Meanwhile, for every one-year increment in the duration of medications ≥240 days, the QOL will increase by 28.9%. Patients who had been on medication for a longer period of time were more knowledgeable about how to use it, resulting in better disease control and subsequently better QOL. This is in contrast to a study by Rawle et al., which found that patients with long-term polypharmacy had stronger negative correlations with cognitive and physical performance [40]. Masnoon et al. also suggested that patients with chronic use of medications, may be at the greatest risk of medication-related issues [9].

The MLR analysis also revealed that the low-income group (B40) has higher QOL. Poorer T2DM control in higher income groups can be explained perhaps due to adopting an unhealthy lifestyle [41]. However, several studies show that lower income status was associated with poorer QOL [42,43].

### 4.4. Strengths and Limitations of the Study

The main strength of this study is the novelty of its findings in demonstrating relationships between SE in MU and QOL among elderly with T2DM on polypharmacy in Malaysian primary care specialist clinic. An additional strength is the utilization of valid and reliable tools which have been validated for the Malaysian population.

Limitations of this study include the non-probability sampling bias. Nonetheless, throughout data collection, efforts were made to enroll all elderly patients with T2DM who were on polypharmacy to participate in this study. This study also selected elderly individuals with T2DM on polypharmacy who received follow up care at the clinic at least once. Findings of this study may not be generalizable to other primary care clinic in urban areas. The fact that this study did not explore at other factors that could influence QOL such as comorbidities (heart failure, obesity) or psychological status (anxiety or depression), the findings of multiple linear regression should be interpreted with caution. Other limitation is that while an association between SE in MU and QOL was found, the causality could not be proven. Information bias may also occur as a result of the fact that only one investigator was trained and involved in the study procedures prior to the study’s conduct in order to reduce variability in the method of data collection.

### 4.5. Implication for Clinical Practice and Future Research

This study shows that better SE in MU are important determinants of good QOL. Findings of this study suggest the importance to include routine use of SE in MU measures in the management of T2DM in primary care aiming to improve QOL. Assessment of SE in MU in patients with T2DM should be an important initial step in developing individually tailored interventions. These interventions should also focus on improving SE in MU to increase QOL. Efforts should be made by primary care providers together with the pharmacist since they are well-positioned to interact with patients about their drugs and can encourage active patient participation in prescription decisions [44]. Thus, this can enhance patients’ SE in order to improve their SE in MU, and ultimately, QOL. Patients may benefit from a routine medication review with their treating physician or pharmacist in order to improve SE in MU, subsequently leading to better QOL. Several measures can be taken to ensure that patients understand what their physicians are saying to them, including open communication, communicating in a simple and understandable language, using only key points, avoiding excessive information and medical jargon, talking slowly, using comparative examples and encouraging patients to ask questions [45].

Primary care practitioners should be encouraged to provide MU support to their patients with T2DM in order to increase SE and QOL. However, more research involving systematic random sampling of patients with T2DM in a larger number of public primary care clinics in Malaysia is needed to confirm the findings of this study.

## 5. Conclusions

In conclusion, this study discovered that SE in MU is the strongest factor contributing to better QOL among our participants, with cautions of unstudied variables such as concomitant co-morbidities, other health factors and health literacy. Despite its limitations, this is the only study that has looked into such relationships among elderly with T2DM on polypharmacy in a Malaysian primary care setting. The findings of this study emphasize the importance of measuring SE in MU in order to improve QOL.

## Figures and Tables

**Figure 1 ijerph-19-03031-f001:**
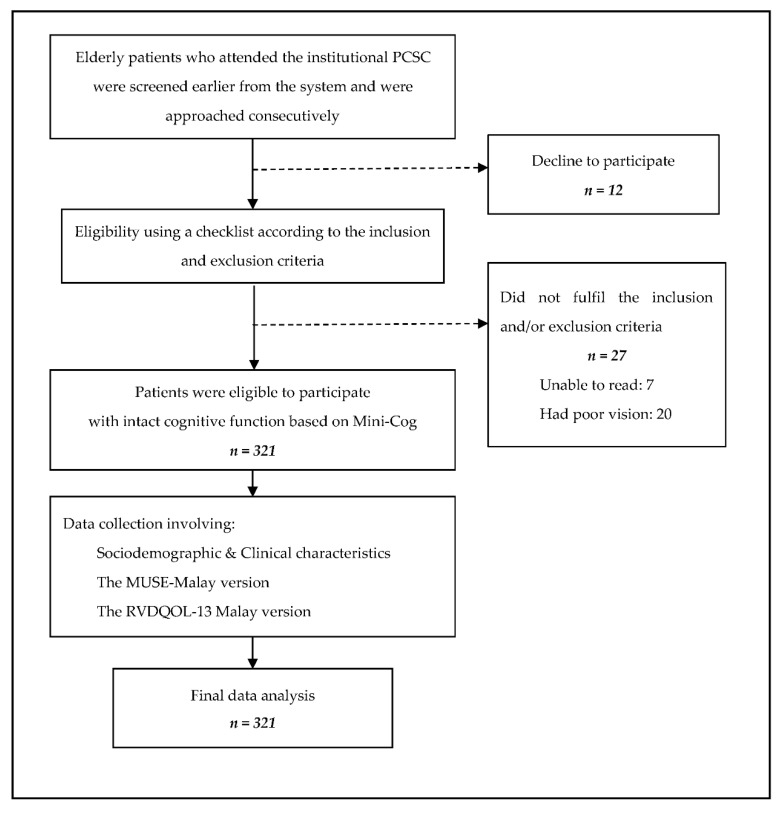
Flow of the conduct of the study.

**Figure 2 ijerph-19-03031-f002:**
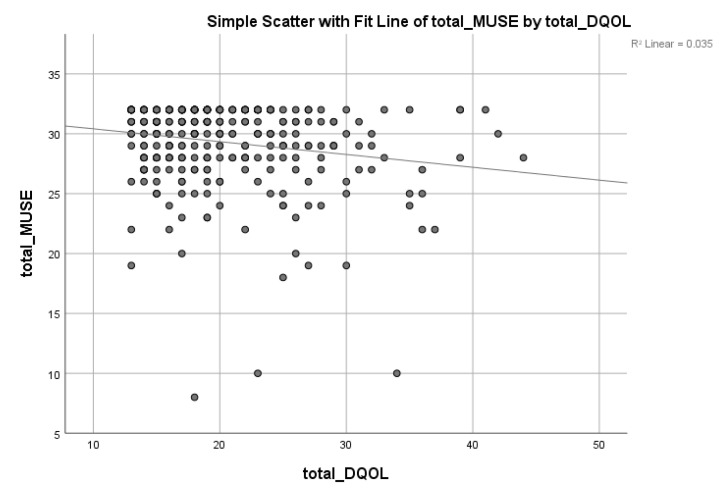
There is a negative rank correlation between total MUSE and total DQOL (*p* = 0.012).

**Table 1 ijerph-19-03031-t001:** Sociodemographic and clinical characteristics of the participants with T2DM on polypharmacy (n = 321).

Variables	n (%)
Age (years) Mean (±SD) 66.7 (±0.286)	
60–69 (young-old)	243 (75.7)
70–79 (middle-old)	73 (22.7)
≥80 (very-old)	5 (1.6)
Gender	
Male	187 (58.3)
Female	134 (41.7)
Ethnicity	
Malay	272 (84.7)
Chinese	22 (6.9)
Indian	26 (8.1)
Others	1 (0.3)
Marital status	
Single	9 (2.8)
Married	259 (80.7)
Divorce	53 (16.5)
Education level	
No formal education	4 (1.2)
Primary	42 (13.1)
Secondary	155 (48.3)
Tertiary	120 (37.4)
Occupation	
Public sector	15 (4.7)
Private sector	13 (4.0)
Self-employed	28 (8.7)
Housewife	61 (19.0)
Pensioner	204 (63.6)
Household income	
B40 (<RM4850)	207 (64.5)
M40 (RM4850-10959)	100 (31.2)
T20 (>RM10960)	14 (4.4)
Number of Medications	
≥5–9	293 (91.3)
≥10	28 (8.7)
Duration of taking medications	
≥90 days–239 days	76 (23.7)
≥240 days	245 (76.3)
Hypertension	
Present	311 (96.9)
Absent	10 (3.1)
Cardiovascular Related Disease	
Present	133 (41.4)
Absent	188 (58.6)
Dyslipidemia	
Present	320 (99.7)
Absent	1 (0.3)
Musculoskeletal Related Disease	
Present	44 (13.7)
Absent	277 (86.3)
Gastrointestinal Related Disease	
Present	30 (9.3)
Absent	291 (90.7)
Respiratory Related Disease	
Present	44 (13.7)
Absent	277 (86.3)
Other Related Disease	
Present	125 (38.9)
Absent	196 (61.1)
Modality of treatment	
Pills only	184 (57.3)
Pills and insulin	108 (33.6)
Pills and others (e.g., MDI)	29 (9.0)

**Table 2 ijerph-19-03031-t002:** The median MUSE subscale scores among elderly with T2DM on polypharmacy (n = 321).

Subscale Domain	Subscale Items	Median (IQR)	Total Score
	Items (Scale 1 = Strongly Disagree, 2 = Slightly Disagree, 3 = Slightly Agree, and 4 = Strongly Agree)		
**Taking medication**	**Item 1: It is easy for me to take my medicine on time**	4.0 (1)	
	Item 6: It is easy to remember to take all my medicines	4.0 (1)	
	Item 7: It is easy for me to set a schedule to take my medicines each day	4.0 (1)	
	Item 8: It is easy for me to take my medicines every day	4.0 (0.5)	
	Total median subscale domain score	4.0 (0.5)	16 (2)
Learning about medication	Item 2: It is easy for me to ask my pharmacist questions about my medicine.	4.0 (1)	
	Item 3: It is easy for me to understand my pharmacist’s instructions for my medicine.	4.0 (0)	
	Item 4: It is easy for me to understand instructions on medicine bottles.	4.0 (0)	
	Item 5: It is easy for me to get all the information I need about my medicine	4.0 (1)	
	Total median subscale domain score	3.75(0.5)	15 (2)
**Total median MUSE score ****		**3.75(0.5)**	**30 (4)**

** higher score (near 4) corresponds with higher comprehension of written prescription instructions.

**Table 3 ijerph-19-03031-t003:** The median DQOL subscale scores among elderly with T2DM on polypharmacy (n = 321).

Subscale Domain	Subscale Items	Median (IQR)	Total Score	Percentage Score (%)
Satisfaction	Items (scale 1 = very satisfied, 2 = moderately satisfied, 3 = neither satisfied nor dissatisfied, 4 = moderately dissatisfied and 5 = very dissatisfied)			
	Item 1: How satisfied are you with the amount of time it takes to manage your diabetes?	1.0 (1)		
	Item 2: How satisfied are you with the amount of time you spend getting check-ups?	1.0 (1)		
	Item 3: How satisfied are you with the time it takes to determine your sugar level?	1.0 (1)		
	Item 4: How satisfied are you with your current treatment	1.0 (0)		
	Item 5: How satisfied are you with your knowledge about your diabetes?	2.0 (1)		
	Item 6: How satisfied are you with life in general?	1.0 (1)		
	Total median subscale domain 1 score	1.33 (0.5)	8 (3)	26.67%
Impact	Items (scale 1 = never, 2 = sometimes, 3 = often, 4 = frequently and 5 = always			
	Item 1: How often do you feel pain associated with the treatment for your diabetes?	1.0 (1)		
	Item 2: How often do you feel physically ill?	2.0 (1)		
	Item 3: How often does your diabetes interfere with your family life?	1.0 (1)		
	Item 4: How often do you find your diabetes limiting your social relationships and friendships?	1.0 (1)		
	Total median subscale domain 2 score	1.5 (1)	6 (4)	30.00%
Worry	Items (scale 1 = never, 2 = sometimes, 3 = often, 4 = frequently and 5 = always			
	Item 1: How often do you worry about whether you will pass out?	1.0 (1)		
	Item 2: How often do you worry that your body looks different because you have diabetes?	1.0 (1)		
	Item 3: How often do you worry that you will get complications from your diabetes?	2.0 (1)		
	Total median subscale domain 3 score	1.66 (1)	5 (3)	33.33%
**Total median DQOL score** **		**1.46 (0.58)**	**19 (8)**	**29.23%**

** Higher score indicates poorer quality of life.

**Table 4 ijerph-19-03031-t004:** Spearman rank correlation between domains of MUSE and DQOL among elderly with T2DM on polypharmacy (n = 321).

Domains	Spearman’s Correlation (*r*)	*p* Value
Taking medication and satisfaction	−0.242	**<0.001** *
Taking medication and impact	−0.136	**0.015** *
Taking medication and worry	−0.096	0.085
Learning medication and satisfaction	−0.228	**<0.001** *
Learning medication and impact	−0.027	0.632
Learning medication and worry	−0.043	0.438
**Total MUSE and DQOL**	**−0.140**	**0.012** *

* Statistical significance at *p* < 0.05; ref-reference group.

**Table 5 ijerph-19-03031-t005:** Simple linear regression to determine the associated factors for DQOL among elderly with T2DM on polypharmacy (n = 321).

	B	SE B	ß	Sig
Constant	20.679	0.387		
Age group				
60–69	1			ref
70–79	−1.706	0.805	−0.118	**0.035**
≥80	0.121	2.726	0.002	0.965
Note: R² = 0.014, * *p* < 0.25				
Constant	20.529	0.367		
Ethnicity				
Malay	1			ref
Chinese	−2.393	1.342	−0.100	**0.075**
Indian	−0.914	1.243	−0.041	0.463
Others	0.471	6.064	0.004	0.938
Note: R² = 0.011, * *p* < 0.25				
Constant	23.889	2.015		
Marital status				
Single	1			ref
Married	−3.719	2.050	−0.243	**0.071**
Divorce	−3.606	2.180	−0.221	**0.099**
Note: R² = 0.01, * *p* < 0.25				
Constant	23.933	1.550		
Occupation				
Public sector	1			ref
Private sector	−4.472	2.275	−0.146	**0.050**
Self-employed	−1.648	1.921	−0.077	0.392
Housewife	−4.114	1.730	−0.267	**0.018**
Others	−3.987	1.606	−0.317	**0.014**
Note: R² = 0.03, * *p* < 0.25				
Constant	22.286	1.595		
Household income				
T20	1			ref
M40	−0.536	1.703	−0.041	0.753
B40	−2.832	1.648	−0.224	**0.087**
Note: R² = 0.035, * *p* < 0.25				
Constant	20.126	0.353		
Number of medications				
≥5–9	1			ref
≥10	1.909	1.195	0.089	**0.111**
Note: R² = 0.08, * *p* < 0.25				
Constant	23.368	0.668		
Duration of taking medications				
90–239 days	1			ref
≥240 days	−4.030	0.764	−0.283	**<0.001**
Note: R² = 0.080, * *p* < 0.25				
Constant	20.622	0.442		
Cardiovascular related disease				
Absent	1			ref
Present	−0.795	0.686	−0.065	**0.247**
Note: R² = 0.004, * *p* < 0.25				
Constant	20.462	0.364		
Respiratory related disease				
Absent	1			ref
Present	−1.235	0.982	−0.07	**0.210**
Note: R² = 0.005, * *p* < 0.25				
Constant	19.969	0.432		
Other Related Disease				
Absent	1			ref
Present	0.831	0.693	0.067	**0.232**
Note: R² = 0.004, * *p* < 0.25				
Constant	19.418	0.439		
Modality of treatment				
Pills only	1			ref
Pills and insulin	2.544	0.722	0.199	**<0.001**
Pills and others	0.202	1.190	0.010	0.865
Note: R² = 0.039, * *p* < 0.25				
Constant	29.548	2.852		
MUSE (domain)				
Taking medication	−0.442	0.191	−0.152	**0.021**
Learning about medication	−0.189	0.214	−0.058	0.378
Note: R² = 0.036, * *p* < 0.25				
Constant	29.804	2.827		
MUSE (total)				
Total MUSE	−0.324	0.96	−0.186	**0.001**
Note: R² = 0.035, * *p* < 0.25				

*significant variable as shown by *p* < 0.25

**Table 6 ijerph-19-03031-t006:** Factors associated with QOL by multiple linear regression among elderly with T2DM on polypharmacy (using total MUSE score).

Variables	Standardized Coefficients Beta (β) (95% CI)	t Statistics	*p* Value²
B40	−0.144 (−3.118, −0.534)	−2.781	0.006
Medications ≥10	0.109 (0.214, 4.462)	2.166	0.031
Duration ≥240 days	−0.282 (−5.438, −2.581)	−5.523	<0.001
Pills and insulin	0.193 (1.206, 3.747)	3.836	<0.001
Total MUSE	−0.282 (−5.438, −2.581)	−4.748	<0.001

R² = 0.212. The model reasonably fits well. Model assumptions are met. There are no interaction and multicollinearity problem.

**Table 7 ijerph-19-03031-t007:** Factors associated with QOL by multiple linear regression among elderly with T2DM on polypharmacy (subgroup analysis of MUSE score).

Variables	Standardized Coefficients Beta (β) (95% CI)	t Statistics	*p* Value²
B40	−0.135 (−2.997, −0.420)	−2.609	0.010
Medications ≥10	0.113 (0.273, 4.572)	2.218	0.027
Duration ≥240 days	−0.275 (−5.365, −2.469)	−5.322	<0.001
Pills and insulin	0.190 (1.152, 3.723)	3.731	<0.001
MUSE Domain taking medication	−0.193 (−0.852, −0.270)	−3.794	<0.001

R² = 0.186. The model reasonably fits well. Model assumptions are met. There are no interaction and multicollinearity problem.

## Data Availability

Data are kept at the Department of Primary Care Medicine, Universiti Teknologi MARA, Selangor, Malaysia. Data will be shared upon request and are subject to the applicable and relevant personal data protection laws and regulations.

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
