# Peer review of "Relationship of Self Efficacy in Medication Understanding with Quality of Life among Elderly with Type 2 Diabetes Mellitus on Polypharmacy in Malaysia"

_ijerph, 2022, doi:10.3390/ijerph19053031_

Round 1

Reviewer 1 Report

The manuscript examines the relationship between Self Efficacy in Medication Understanding with Quality of Life using standard questionnaires. The study population of interest is older individuals living in Malaysia (Klang Valley) with Type 2 Diabetes Mellitus and taking more than five medicines everyday. The manuscript is well written, detailed and structured well. The study provides details on the sample size calculations. The manuscript could benefit from minor English language checks. A few other minor issues with the manuscript are:

  1. I suggest that the title be modified to reflect that the study was conducted in Malaysia.
  2. The abstract has too many abbreviations and at times it is difficult to keep up with the multiple use of abbreviations. Please decrease the number of abbreviations to make the abstract more readable.
  3. Lines 108 and 117- please provide references
  4. Line 75- 576 individuals out of how many?

The paper is well written and easy to follow. The study aims to evaluate the relationship of self efficacy in medication understanding with quality of life in older individuals with type 2 diabetes living in Malaysia. The Introduction is detailed and successfully conveys the need for this study and the importance of self efficacy in medication understanding and its effect on quality of life. The introduction references articles from the literature to help establish the research gap and the need for this study well. The tools/ questionnaires used in this study are standard questionnaires used to measure constructs in this population. The sample size is justified scientifically using power analysis. The study uses a large sample (n=321) of older adults with type 2 diabetes taking atleast 5 medicines everyday. Appropriate statistical analysis is conducted and the conclusions reflect the study results correctly. The discussion also details the strengths and weaknesses of the study.

Author Response

Dear Reviewer 1

Reviewer 1

Reviewer’s Comments

Authors’ Responses

Overall

 The manuscript examines the relationship between Self Efficacy in Medication Understanding with Quality of Life using standard questionnaires. The study population of interest is older individuals living in Malaysia (Klang Valley) with Type 2 Diabetes Mellitus and taking more than five medicines every day. The manuscript is well written, detailed and structured well. The study provides details on the sample size calculations.

The paper is well written and easy to follow. The study aims to evaluate the relationship of self-efficacy in medication understanding with quality of life in older individuals with type 2 diabetes living in Malaysia. The Introduction is detailed and successfully conveys the need for this study and the importance of self-efficacy in medication understanding and its effect on quality of life. The introduction references articles from the literature to help establish the research gap and the need for this study well. The tools/ questionnaires used in this study are standard questionnaires used to measure constructs in this population. The sample size is justified scientifically using power analysis. The study uses a large sample (n=321) of older adults with type 2 diabetes taking at least 5 medicines every day. Appropriate statistical analysis is conducted and the conclusions reflect the study results correctly. The discussion also details the strengths and weaknesses of the study.

Thank you very much for your kind comments.

Comment 1:

The manuscript could benefit from minor English language checks. A few other minor issues with the manuscript are:

I suggest that the title be modified to reflect that the study was conducted in Malaysia.

Thank you for your opinion.

The manuscript has undergone proofreading.

The revised title is “Relationship of Self Efficacy in Medication Understanding with Quality of Life Among Elderly with Type 2 Diabetes Mellitus on Polypharmacy in Malaysia”.

Comment 2:

The abstract has too many abbreviations and at times it is difficult to keep up with the multiple use of abbreviations. Please decrease the number of abbreviations to make the abstract more readable.

Thank you for the recommendation.

We have reduced the number of abbreviations in response to suggestions.

The changes can be seen on page 1.

Comment 3:

Lines 108 and 117- please provide references

Thank you for the suggestion.

We have provided the reference.

The changes can be seen on page 3, line 106.

Comment 4:

Line 75- 576 individuals out of how many?

Thank you for the suggestion.

We have included the number.

‘Polypharmacy was found to be 45.9% common among urban community-dwelling elderly in Malaysia, with 576 people out of 1256 are using at least five drugs.’

The changes can be seen on page 2, line 74.

Reviewer 2 Report

The authors aimed to determine the relationship between Self-efficacy (SE) in Medication understanding (MU) and quality of life (QOL), and the factors associated with QOL in elderly with type 2 diabetes mellitus (T2DM) on polypharmacy. This cross-sectional study was conducted on these populations at primary care specialist clinic.

The study covers some issues that have been overlooked in other similar topics. The structure of the manuscript appears adequate and well divided in the sections. Moreover, the study is easy to follow, but few issues should be improved. Some of the comments that would improve the overall quality of the study are:

1-) The manuscript needs grammar correction. Please also check typos thorough the text;

2-) Conclusion Section: This paragraph required a general revision to eliminate redundant sentences and to add some "take-home message".

Author Response

Dear Reviewer 2

Reviewer 2

Reviewer’s Comments

Authors’ Responses

Overall

The authors aimed to determine the relationship between Self-efficacy (SE) in Medication understanding (MU) and quality of life (QOL), and the factors associated with QOL in elderly with type 2 diabetes mellitus (T2DM) on polypharmacy. This cross-sectional study was conducted on these populations at primary care specialist clinic.

The study covers some issues that have been overlooked in other similar topics. The structure of the manuscript appears adequate and well divided in the sections. Moreover, the study is easy to follow, but few issues should be improved.

Thank you for your kind suggestions.

Comment 1:

Some of the comments that would improve the overall quality of the study are:

The manuscript needs grammar correction. Please also check typos thorough the text.

Thank you for your comments.

The manuscript has undergone proofreading.

Comment 2:

Conclusion Section: This paragraph required a general revision to eliminate redundant sentences and to add some "take-home message".

Thank you for your opinion.

In conclusion, this study discovered that SE in MU is the strongest factor contributing to better QOL among our participants, with cautions of unstudied variables such as concomitant co-morbidities, other health factors and health literacy. Despite its limitations, this is the only study that has looked into such relationships among elderly with T2DM on polypharmacy in a Malaysian primary care setting. The findings of this study emphasize the importance of measuring SE in MU in order to improve QOL.

The changes can be seen on page 15, line 472-477.

Reviewer 3 Report

I was very pleased to read this thorough and interesting paper.

The study is well conducted and the number of patients seems appropriated. I have been very pleased to see a correct use of the Cronbach's α, indeed.

English is fine with only minor spelling mistakes.

Among them:

row 73: involving should be substituted with involve;

row 334-335 (inbetween) Tabel 4: asterisks should be added to p values 0.015 (taking medication and impact) and 0.012 (total MUSE and DQOL)

row 420: some words have been forgotten...

Other than those minor revision requests, I am fine with this paper.

Author Response

Dear Reviewer 3

Reviewer 3

Reviewer’s Comments

Authors’ Responses

Overall

 I was very pleased to read this thorough and interesting paper.

The study is well conducted and the number of patients seems appropriated. I have been very pleased to see a correct use of the Cronbach's α, indeed.

English is fine with only minor spelling mistakes.

Thank you very much for your kind comment.

Comment 1:

Row 73: involving should be substituted with involve

Thank you for noticing the typographical and grammatical errors.

‘Other definitions of polypharmacy involve duration of therapy either 90 days to 239 days or ≥240 days or more (‘long term use’)’

The changes can be seen on page 2, line 72.

Comment 2:

Row 334-335 (in-between) Table 4: asterisks should be added to p values 0.015 (taking medication and impact) and 0.012 (total MUSE and DQOL)

Thank you for the suggestion.

The changes can be seen on page 10.

Comment 3:

Row 420: some words have been forgotten.

Other than those minor revision requests, I am fine with this paper.

Thank you for the suggestion.

The changes can be seen on page 14.

Reviewer 4 Report

This study analyzes the possible relation between x Self-efficacy 17 in medication understanding and quality of life in old patients with Type 2 Diabetes.

This is a well-written study with an original hypothesis trying to confirm the association between self-efficacy in medication understanding and QOL.

The introduction is a little bit long so it should be shortened.

There are three main limitations in this study and they should be mentioned in the discussion, the first one is that it did not explore other factors that could influence QOL like comorbidities (heart failure, obesity) or psychological status (anxiety or depression).

The second limitation is that correlation doesn´t mean causality, so it is no clear if SE is improved, QOL will improve.

Third, the fact that only one investigator was trained and involved in the study procedures before the conduct of the study to minimize variability in the method of data collection. This could be also an information biass.

Minor changes:

  • Lines 157 and 158 are redundant with previous paragraphs.
  • In Table 1, mean age should be removed (removing that column).
  • Maybe Table 2 and 3 could be removed, because they show little information that could be explained in the text briefly.

Author Response

Dear Reviewer 4
Reviewer 4

Reviewer’s Comments

Authors’ Responses

Overall

This study analyses the possible relation between Self-efficacy in medication understanding and quality of life in old patients with Type 2 Diabetes.

This is a well-written study with an original hypothesis trying to confirm the association between self-efficacy in medication understanding and QOL.

Thank you for your kind comment.

Comment 1:

The introduction is a little bit long so it should be shortened.

Thank you for the suggestion.

The authors have summarized the introduction.

Comment 2:

There are three main limitations in this study and they should be mentioned in the discussion, the first one is that it did not explore other factors that could influence QOL like comorbidities (heart failure, obesity) or psychological status (anxiety or depression). The second limitation is that correlation doesn´t mean causality, so it is no clear if SE is improved, QOL will improve. Third, the fact that only one investigator was trained and involved in the study procedures before the conduct of the study to minimize variability in the method of data collection. This could be also an information bias.

Thank you for the suggestion.

We have included the recommendation as below:

‘The fact that this study did not explore at other factors that could influence QOL such as comorbidities (heart failure, obesity) or psychological status (anxiety or depression), the findings of multiple linear regression should be interpreted with caution. Other limitation is that while an association between SE in MU and QOL was found, the causality could not be proven. Information bias may also occur as a result of the fact that only one investigator was trained and involved in the study procedures prior to the study's conduct in order to reduce variability in the method of data collection.’

The changes can be seen on page 14.

Comment 3:

Minor changes:

Lines 157 and 158 are redundant with previous paragraphs.

Thank you for the suggestion.

We have revised the paragraph.

The changes can be seen on page 4, line 155.

Comment 4:

In Table 1, mean age should be removed (removing that column).

Thank you for the suggestion.

The changes can be seen on page 7.

Comment 5:

Maybe Table 2 and 3 could be removed, because they show little information that could be explained in the text briefly.

Thank you for the suggestion.

The authors have decided to maintain the current format, as we believe that it delivers the message clearly.
